# Advances in Portable Heavy Metal Ion Sensors

**DOI:** 10.3390/s23084125

**Published:** 2023-04-20

**Authors:** Tao Hu, Qingteng Lai, Wen Fan, Yanke Zhang, Zhengchun Liu

**Affiliations:** Department of Electronics, School of Physics and Electronics, Central South University, Changsha 410083, China

**Keywords:** heavy metal ions, portable sensing, optical method, electrochemical method

## Abstract

Heavy metal ions, one of the major pollutants in the environment, exhibit non-degradable and bio-chain accumulation characteristics, seriously damage the environment, and threaten human health. Traditional heavy metal ion detection methods often require complex and expensive instruments, professional operation, tedious sample preparation, high requirements for laboratory conditions, and operator professionalism, and they cannot be widely used in the field for real-time and rapid detection. Therefore, developing portable, highly sensitive, selective, and economical sensors is necessary for the detection of toxic metal ions in the field. This paper presents portable sensing based on optical and electrochemical methods for the in situ detection of trace heavy metal ions. Progress in research on portable sensor devices based on fluorescence, colorimetric, portable surface Raman enhancement, plasmon resonance, and various electrical parameter analysis principles is highlighted, and the characteristics of the detection limits, linear detection ranges, and stability of the various sensing methods are analyzed. Accordingly, this review provides a reference for the design of portable heavy metal ion sensing.

## 1. Introduction

With the rapid development of modern agriculture and industry, the environmental problems caused by heavy metal ions are becoming increasingly prominent [1,2,3]. Most heavy metal ions are highly toxic to animals, plants, and humans, even at very low concentrations [4]. In recent years, increasing amounts of heavy metal toxins have been enriched in the environment due to their characteristic degradation resistance. They affect both terrestrial and aquatic communities [5] through the food chain. High concentrations of heavy metals can lead to various health problems, such as triggering the gastrointestinal system [6], kidneys [7], and related neurological [8] diseases. Therefore, trace detection of heavy metal ions in food, soil, and water is vital.

Conventional methods for the detection of heavy metal ions usually include atomic absorption spectroscopy (AAS) [9,10], inductively-coupled plasma/atomic emission spectroscopy (ICP-AES) [11], and inductively-coupled plasma mass spectrometry (ICP-MS) [12]. These techniques have low detection limits and can measure multiple ions simultaneously. However, these techniques generally require specialized personnel and complex sample pre-treatment procedures [13,14]. They also entail large and expensive equipment that is unsuitable for in situ measurements [15,16,17,18,19]. Therefore, portability, low cost, and high integration of detection are important directions for heavy metal ion sensing.

Both optical and electrochemical sensors are highly sensitive and selective in the detection of heavy metal ions. Optical sensors are based on changes in the optical signal caused by the interaction between the recognition element and the target molecule to generate information on the concentration of the target molecule. Different types of optical sensors include fluorescence spectroscopy, colorimetric spectroscopy, Raman spectroscopy, chemiluminescence spectroscopy, infrared spectroscopy, and refractive index spectroscopy sensing [20]. In recent years, optical sensing has been used for the construction of portable heavy metal ion sensors because of its low detection cost, fast response time, and simplicity of operation [21,22,23]. Electrochemical sensors involve an analytical method that converts information about the concentration of a target molecule into the intensity of an electrical signal. In the detection process, the solution to be measured is commonly used as part of a chemical cell, and the change in the electrical parameters of the chemical cell reflects the concentration of the focal substance [24]. With the development of photolithography and 3D printing technology, microelectrodes, such as screen-printed electrodes [25] and fork-finger electrodes [26], have been employed for electrochemical detection and can be combined with analytical circuit and communication modules to achieve a high level of sensing device integration and miniaturization.

The New Era portable ion detection system combines various novel devices and materials based on traditional detection. The addition of these devices and materials not only ensures the accuracy and sensitivity of detection but also takes into account the advantages of low cost, simplicity, efficiency, and miniaturization [27]. The detection system comprises three modules for signal generation, transmission, amplification, and signal reception and processing. In general, each of these parts can be optimized. The use of high quantum yield, and long fluorescence life materials, such as quantum dots, generates stable sensing signals. The combination of smartphones, photometers, and microcontrollers enables visual inspection and wireless data transmission and increases the integration of the detection system. The combination of microfluidic platforms, which integrate the detection process onto a chip, enables the separation of complex samples and avoids the tedious steps involved in traditional detection techniques [28]. These combinations provide practical examples and ideas for the design of portable sensing for field testing.

This paper analyzes the different principles of heavy metal ion detection, including optical and electrochemical methods, the construction of related portable sensors, and the application of various types of sensing in agriculture [29,30], life sciences [31,32], and clinical diagnostics [33,34].

## 2. Optical Sensors

Detection systems using optical principles for ion sensing have become widely used. Optical detection typically relies on the color change resulting from the interaction or reaction between the focal object and the detection reagent [5,21,35]. Optical sensors offer the advantages of simplicity and low cost. At the same time, there is no direct contact between the sensor and the sample during analysis, thereby minimizing the effects of contamination of the sensing probe [36]. The portable optical sensors highlighted in this section include fluorescent, colorimetric, Raman scattering, surface plasmon resonance (SPR), and localized surface plasmon resonance (LSPR) sensors.

### 2.1. Portable Fluorescence Sensing

The basic principle of fluorescent sensors involves analyzing the content of the substance to be measured according to changes in the physicochemical properties of the fluorescent groups after the reaction between the fluorescent molecules and the detection target. These properties include fluorescence intensity and lifetime and are mainly related to charge transfer and energy transfer processes [37,38]. Typically, fluorescent sensors consist of fluorescent and receptor elements that bind specifically to the focal ion. These sensors have received considerable attention because of their low cost, fast detection, wide response range, and simplicity of operation [5,28,39].

Gil et al. [40] performed the fluorescence determination of Hg^2+^ in water and fish samples using a rhodamine 6G derivative-based chemical dosimeter (FC1) and a portable fiber optic fluorescence spectrophotometer. FC1 showed a high molar absorption coefficient and quantum yield, producing strong fluorescence emission at 555 nm, with a fluorescence intensity proportional to the amount of Hg^2+^ at the ng/mL level and a linear detection range of 0 to 12 ng/mL. The portable detection instrument consists of two optical fibers, a charge-coupled device (CCD) camera as the detector, collimating and focusing mirrors, a 500 lm diffraction grating, and an excitation source LED emitting 515 nm radiation with a filter, and AvaSoft 2048 software to control the instrument and correlate the captured fluorescence with the concentration.

Li et al. [41] designed a smartphone-based three-channel ratiometric fluorescence device for high-sensitivity sensing of Hg^2+^, Fe^3+^, and Cu^2+^ with detection limits of 3, 0.5, and 30 nM, respectively, according to the fluorescence burst mechanism of the three metal ions on three doped carbon quantum dots. Their proposed device is portable for field analysis use. The device also includes a rechargeable power supply, UV light source, reaction cup, shading box, reflector, and smartphone signal acquisition platform. This design eliminates the influence of light source intensity and ambient temperature on the fluorescence signal and improves the accuracy and reliability of the detection results.

The choice of non-toxic sensor materials is essential to avoid secondary contamination from toxic fluorescent dyes in field detection. Nath et al. [42] developed a paper-based integrated sensor device for simple Pb^2+^ and Cu^2+^ ion detection. Using lipoic acid solution as a probe to which the Pb^2+^ or Cu^2+^ to be measured is added, the fluorescence intensity of the solution decreases, and its color changes from red to blue because of the aggregation of gold nanoparticles (AuNPs). The authors achieved the detection of 1 ppb Pb^2+^ or Cu^2+^ on a Y-shaped test strip, where the Pb^2+^ or Cu^2+^ and the lipoic acid solutions were able to mix and interact sufficiently to produce a more pronounced signal.

To overcome the poor optical stability of fluorescent dyes, semiconductor quantum dots with better optical stability, high quantum yields, and long fluorescence lifetimes have been used for ion sensing. Chen et al. [43] investigated green, orange, and red luminescent quantum dots of CdTe covered with thioglycolic acid (TGA). Given the passivation and binding effect of Ag^+^ on the quantum dots, the transfer of electrons from the dots to Ag^+^ induces a red shift in the fluorescence spectrum and a reduction in fluorescence intensity, enabling the detection of trace amounts of Ag^+^. Accordingly, the authors designed a homemade portable sensing device (Figure 1a) for in situ detection. The detection system uses a 32-bit embedded microprocessor unit as the control center to convert the fluorescence signal after excitation of the UV photodiode into an electrical signal and establish a linear relationship between the voltage signal and the Ag^+^ concentration. The portable system has a detection limit of 5 nM and a linear detection range of 5 to 200 nM.

Fluorescence resonance energy transfer (FRET) is an energy transfer phenomenon between two fluorescent molecules in close proximity [46]. FRET can also be used for heavy metal ion detection. Sub-nanosecond anisotropic and non-linear fluorescence decay from molecular rotation are the main challenges for FRET detection, and these issues are addressed by the pulse duration and stability of laser light sources. Zhou et al. [44] developed a portable evanescent wave optofluidic biosensor (EWOB) with a system that uses fluorescently labeled poly-A DNA strands (CY-A14) and burster-labeled poly-T DNA strands (BQ-T14) for the highly sensitive detection of Hg^2+^, with a detection limit of 8.5 nM and a detection time of less than 10 min. In this platform (Figure 1b), laser light is emitted through a single multimode fiber optic coupler (SMFC) into a tapered fiber optic probe. The incident light is transmitted through the fiber probe by total internal reflection, and the electromagnetic field extending from the fiber probe interface into the solution is the evanescent wave [47]. The higher the Hg^2+^ concentration, the less CY-A14 is bound to BQ-T14, enabling the freer CY-A14 to be excited by the evanescent wave generated on the fiber optic probe surface, the stronger the fluorescence intensity detected by the EWOB.

Wang et al. [45] developed a novel portable whole-cell biosensing platform by integrating a simple handheld fiber-optic dissolved oxygen sensor and bacterial cultures or lyophilized bacteria. The addition of heavy metal ions inhibited *E. coli* respiration, enabling rapid detection of the acute toxicity of heavy metal ions. In Figure 1c, a multimode fiber optic bundle transmits excitation light and collects fluorescence, which greatly simplifies the structure of the dissolved oxygen sensor and improves the efficiency of light transmission. An O_2_ sensing foil containing a fluorescent oxygen-sensitive probe was applied to the end of the fiber bundle to construct the fiber node. Such an optical design achieves high portability and stability as it does not require other optical separation elements. Under optimal conditions, the detection limit and IC50 (semi-inhibitory concentration) of *E. coli* cultures for Hg^2+^ were 5.62 and 11.64 μM, respectively. 

### 2.2. Portable Colorimetric Sensors

Colorimetry is a method of measuring the content of a target by comparing or calculating the color depth of colored solutions, relying on Lambert’s law and based on the production of colored compounds [48,49]. The colorimetric method provides a clear color change that can generally be identified with the naked eye. Given the simplicity and straightforwardness of colorimetric instruments, many point-of-care testing (POCT) systems for the immediate detection of non-conventional heavy metal ions are based on the colorimetric method [27,28].

Portable colorimetric sensing emphasizes smartphone integration. Xiao et al. [50] presented a simple, sensitive, and portable Hg^2+^ detection system based on a smartphone and nano-adaptor colorimetric sensor with a detection time of only 20 min. A smartphone equipped with a photometer application captures and processes the signal from the microporous reader on a smartphone (MR S-phone). The system provides linear colorimetric readings of Hg^2+^ concentrations in the range of 1 to 32 ng/mL, with a correlation coefficient of 0.991 and a limit of detection (LOD) of 0.28 ng/mL for Hg^2+^. Li et al. [51] developed a smartphone-based 3D printed sensing device (Figure 2a) for fast and reliable colorimetric detection. Optical sensors and light meter applications built into smartphones are used to read the test results with detection limits of 2.18 × 10^−2^ μg/mL, 1 μg/mL, 0.001 μg/mL, and 0.02 μg/mL for Pb^2+^, Hg^2+^, Cd^2+^, and Zn^2+^, respectively. In addition, the reagents are highly resistant to interference and are suitable for detection in real water samples. 

Combining a smartphone with fluorescent paper strips allows instantaneous semi-quantitative detection of metal ions, significantly reducing detection time and costs. Wang et al. [55] developed an effective colorimetric fluorescence detection method for lead ions in water by printing paper strips with a fluorescent probe solution and using a smartphone application (color recognition) to achieve a visual, real-time, semi-quantitative detection, with a detection limit of 2.89 nM. When the test strip was immersed in the sample solution, the blue fluorescence was burst by the Pb^2+^ in the solution, and the red fluorescence, as a background reference, remained unchanged under UV light. A significant color change from blue to red was observed, resulting in a semi-quantitative visual detection. Subsequently, by identifying the RGB values of the fluorescent probe solution and the corresponding paper strips, the smartphone allowed the visual detection of Pb ions. Firdaus et al. [52] investigated a low-cost, simple, and portable method for quantitative analysis of Hg^2+^ based on digital image colorimetry combined with smartphone applications (Figure 2b). A small number of silver nanoparticles (AgNPs) was used as a colorimetric agent with high selectivity for Hg^2+^. The brownish-yellow AgNPs instantly became colorless after the addition of Hg^2+^ to the redox reaction. Firdaus et al. not only attached AgNPs to the media to create a paper-based analytical device but also included an Android app in the Google Play Store for data processing. The method has a detection limit of 0.86 ppb, which is comparable to the sensitivity of large conventional instruments.

The combination of a colorimetric sensing method with a photometer also enables the miniaturization of the detection device. Yu et al. [53] designed a portable Cr^3+^ assay system based on a smartphone readout device and an enzyme-linked immune-sorbent assay (ELISA) (Figure 2c). The readout device consisted of a light source and a miniaturized detection platform, and the mobile phone application “Photometer” was used to collect and process the signal from the readout device. The signal acquired by the new system is positively correlated with the Cr^3+^ concentration (R^2^ = 0.995), with a linear detection range of 0.8 to 50 ng/mL and a detection limit of 0.81 ng/mL. The system showed good selectivity in the detection of Cr^3+^. Camilo et al. [56] produced a micro-controlled photometer based on light-emitting diodes (led) using AuNPs for the detection of Pb^2+^. The principle of measuring Pb^2+^ is based on the color change in AuNPs after aggregation caused by Pb^2+^. The photometer uses a single LED as the light source, the sensor TCS230, an Arduino electronic card as the acquisition system, and software written in C++ to control the photometer and perform data acquisition. This system has a LOD of 0.89 mM and shows excellent selectivity for Pb^2+^.

In response to the low integration of the device and the low level of automation and visualization, Zhao et al. [54] proposed a portable analytical system based on a AuNPs probe and chip laboratory (Figure 2d). A custom microplate and a handheld colorimetric reader were designed for the colorimetric detection of Pb^2+^ and Al^3+^ in water, displaying the detection data on an integrated LCD and enabling wireless transmission of data to other devices. Calibration experiments have shown that the system achieves detection limits of 30 ppb for Pb^2+^ and 89 ppb for Al^3+^, both of which are comparable to benchtop analytical spectrometers. The colorimetric readout consists of 8 × 8 bi-color LEDs, which emit light at wavelengths of 512 to 518 nm (green) and 610 to 625 nm (red), and the microprocessor controls the emission wavelength by varying the supply voltage of the LED array. The voltage signal from the photodiode is transmitted via USB to a computer for data analysis.

### 2.3. Portable Raman Scattering Sensors

Raman spectroscopy identifies molecules by fingerprinting information from molecular vibrational spectra in analytical chemistry [57], biochemical sensing [58], and environmental monitoring [59]. Portable Raman scattering sensing has become an important tool for trace monitoring of environmental components and biochemical sensing because of its unique advantages of high sensitivity, unique spectral fingerprinting, and non-destructive data acquisition [60].

Portable Raman scattering sensing is often combined with microfluidic devices for the detection of heavy metal ions, thereby avoiding problems such as variable mixing times, scattering geometry, local heating, and photodissociation in conventional surface-enhanced Raman spectroscopy (SERS) detection under static conditions. Qi et al. [61] combined SERS with a microfluidic platform to achieve rapid quantitative detection of As^3+^ ions. Using AgNPs as a SERS-enhanced substrate, glutathione (GSH) and 4-mercaptopyridine (4-MPY) are bound to the surface of the AgNPs. When As^3+^ ions encounter GSH/4-MPY-functionalized AgNPs, the initially dispersed probe aggregates because of the stronger affinity of As^3+^ ions for GSH. Consequently, the Raman signal of 4-MPY adsorbed on the surface of AgNPs is enhanced, and As^3+^ ions are detected accordingly. Qi et al. designed a zigzag PDMS microfluidic channel to allow the efficient and rapid mixing of the two confluent streams, with a channel width of 350 μM and a depth of 50 μM. The linear range for the quantitative analysis of this As^3+^ ion sensing was 3–200 ppb, with a detection limit of 0.67 ppb.

However, this type of sensing generally requires in situ binding of the active material to the microfluidic channel, yielding a disposable sensor. To overcome this limitation, conductor-precious metal nanocomposites were used to enhance the SERS effect. He et al. [62] developed nucleic acid aptamer-modified sea urchin ZnO-Ag arrays that could be reused more than three times for the construction of rigid SERS biochips. In the absence of UO_2_^2+^, the rhodamine B-labeled double-stranded DNA formed a rigid structure. Only a weak Raman signal was detected, but the Raman signal was amplified by pumping the UO_2_^2+^ solution into the microfluidic device (Figure 3). Thus, the device can be used for ultra-sensitive and efficient detection of UO_2_^2+^ ions in real environments with a detection limit of 7.2 × 10^−7^ μM. Similarly, to ensure the reproducibility and stability of the SERS intensity during the detection process, Zhang et al. [63] demonstrated an I^−^-functionalized SERS substrate, as I^−^ can enhance SERS intensity by co-adsorption with a positively charged Raman reporter molecule (crystalline violet), for rapid and sensitive detection of Hg^2+^ on a highly integrated microfluidic platform. The sealed nature of the microfluidic device avoids interference from the environment and further improves detection efficiency and accuracy.

### 2.4. Local Surface Plasmon Resonance Sensing (LSPR/SPR)

The mechanism of the SPR sensor is founded on the frequency sensitivity of the oscillating electrons to the plasma nanoparticle environment [64,65]. If an electromagnetic wave (e.g., light) incident on a metal nanoparticle has a wavelength much higher than the size of the nanoparticle, then the conduction electrons will collectively begin to oscillate at a specific frequency, leading to the phenomenon known as SPR. When this collective oscillation occurs on a finite volume of particulate matter, LSPR occurs [66]. Both methods can be used for the rapid, accurate, and specific detection of heavy metal ions in water.

Recently, functionalized AuNPs have been used in SPR sensors, where their high refractive index and strong plasmonic absorption properties can improve detection sensitivity. Yuan et al. [67] developed a fiber-optic sensor for the portable and inexpensive detection of Hg^2+^ based on the SPR effect. They used 4-mercaptopyridine (4MPY) functionalized AuNPs (AuNPs/4-MPY) as signal amplification markers (Figure 4a), with a detection limit of 8 nM for Hg^2+^, under optimal conditions. Dhara et al. [68] used a reflection-based LSPR fiber-optic sensor to detect the concentration of Pb^2+^ in aqueous solutions, determined by a link between the binding rate of functionalized AuNPs immobilized on the fiber surface to Pb^2+^ ions and the shift of the LSPR resonance wavelength. The broadband tungsten light source employed was connected to a fiber optic coupler, and the other two ends of the fiber optic coupler were connected to the spectrophotometer and the AuNP-coated fiber. Due to the silver coating on the tip of the fiber coated with gold nanoparticles, the spectrophotometer is able to collect reflected light (Figure 4b). A PC was utilized to monitor and record the real-time absorption spectra observed in the wavelength range of 400 to 900 nm, with a sensitivity of 0.28 nm/mM for Pb^2+^ ions and a response time of 30 s.

Fiber optic plasmon resonance (FO-SPR) offers the advantages of lower cost and portability compared to classical SPR devices. BG et al. [69] prepared a gold-plated reflective fiber-optic-surface plasmon resonance sensor, functionalized with bovine serum albumin, for the detection of cadmium ions, with a detection limit of 7.1 nM and a sensitivity of 76.67 nm/μM. The FO-SPR portable sensing system (Figure 4c) consists of a UV-Vis spectrophotometer, a tungsten halogen lamp source, a bifurcated FO, and an interchangeable FO-SPR sensor. The spectrophotometer is further connected to a laptop computer for measuring light reflected by the FO sensing head.

To achieve lower detection limits, the temperature self-compensation capability and high refractive index sensitivity of high-performance tilted fiber Bragg grating (TFBG-SPR) sensors can be exploited. Wang et al. [70] designed a new DNAzyme biosensor for Pb^2+^ detection based on the “hot spot” effect in the near-infrared band using a compact, high-performance TFBG-SPR sensing platform. The device takes advantage of the specific catalytic reaction of DNAzyme with Pb^2+^ and the “hot spot” effect of excitation by a narrow gap between the falling AuNPs and the metal membrane to achieve Pb^2+^ detection. To evaluate the utility of the TFBG-SPR biosensor, Pb^2+^ was added to various concentrations of tap water and clinical human serum samples. The fiber-optic biosensor was able to detect as low as 8.56 pM while obtaining a large dynamic response range of 10^−11^ M to 10^−6^ M.

## 3. Electrochemical Sensors

As the demand for heavy metal detection continues to grow, electrochemical detection is favored for its high sensitivity and efficiency [71,72]. The basic principle of electrochemical sensors involves using a constant potentiometer to output a transducer signal and identify the potential difference, as the presence of heavy metal ions can cause changes in various electrical parameters such as voltage, potential, impedance, conductance, and capacitance [73]. Therefore, the main techniques applied to electrochemical detection are voltammetry, impedance, potentiometry, and conductivity.

### 3.1. Voltammetry

Voltammetric techniques are widely used for the detection of heavy metal ions in a variety of complex environments. One of the effective voltammetric methods is anodic stripping voltammetry (ASV), which offers the advantages of high sensitivity and a wide linear dynamic range. Katiyar et al. [74] designed a low-cost electronic circuit for the detection of Cd and Pb ions in soil samples (Figure 5a). They connected screen-printed electrodes and glassy carbon electrodes to a circuit designed for electrochemical analysis to detect ion concentrations, according to the ASV principle. To make the system portable, a voltage control circuit was used to implement the electrochemical laboratory (Ec-Lab) function.

Square wave voltammetry (SWV) is also widely used in portable detection. Li et al. [75] developed a portable electrochemical sensor system for the detection of Cu^2+^ in water samples (Figure 5b). The sensor hardware consists of a custom electrochemical electrode and a miniaturized detection circuit module. The detection circuit module uses an ARM chip to achieve precise control of the multi-channel constant potential meter and the acquisition of weak current signals, and the detection results are transferred to an Android PAD via Bluetooth. The SWV test results showed that the sensor had a sensitivity of 0.0075 μA/μgL^−1^ for Cu^2+^ in the 0 to 400 μg/L concentration range. Gao et al. [76] designed a wearable microsensor array to simultaneously monitor Zn, Cd, Pb, Cu, and Hg ions in human body fluids by electrochemical SWASV on Au and Bi microelectrodes. The sensor array is encapsulated in a wristband (Figure 5c). Cu and Zn in skin sweat were monitored through the band by using an electrochemical constant potential meter. The oxidation peaks of these metals were calibrated and compensated by incorporating a skin temperature sensor, allowing detection limits of up to 200 μg/L. Their research provides a reference for the implementation of non-invasive wearable portable sensing.

Pawar et al. [78] proposed a new method for sensing and detecting heavy metals based on the memristor switching effect to investigate the fingerprinting characteristics of memristor-based devices: hysteresis loop and limiting linearity in the current–voltage (I-V) plane. A memristor, which is a circuit device that represents the relationship between magnetic flux and electrical charge, has a resistance value determined by the electrical charge flowing through it [79]. By measuring the resistance of a memristor, the amount of charge flowing through it can be identified and thus has the effect of remembering the charge. Consequently, the memristor effect can be used to develop high-performance sensors [80]. The system designed by Pawar et al. showed current–voltage signals for all metal ions tested, with detection limits as low as 0.1 to 10 ppm for Cd^2+^ ions and is expected to be used in portable devices.

### 3.2. Impedance Method

Some of the most widely used impedance measurement techniques for determining the concentration of analytes in aqueous solutions include electrochemical impedance spectroscopy (EIS) and alternating current voltammetry [81]. Of these two techniques, EIS is often applied for the analysis of metal ions in different biological and other environmental samples. Wang et al. [77] developed a rapid screening tool for Pb ions in blood samples. The test system consisted of an impedance-modulated field effect transistor (FET) sensor based on an ion-selective membrane and a portable sensor measurement unit (Figure 5d). The impedance change induced by Pb ions is amplified by the FET, resulting in a detection limit close to 10^−5^ μM and a detection time of 15 min. This handheld system can be easily operated. 

Gupta et al. [82] developed an electrochemical label-free portable impedance sensor for the detection of Pb^2+^ in water using α-MnO_2_/GQD as the acceptor. A proportional relationship between sensor resistance and Pb^2+^ concentration was established for this portable sensor, and the detection results showed significant linearity over a wide linear response range of 0.001 nM to 1 M. The LOD and sensitivity of the developed sensor were calculated to be 0.81 nM and 1.05 kω/nM/mm^2^, respectively. Radovanovic et al. [26] produced a portable sensor consisting of an impedance analyzer connected to a computer and a multilayer sensor platform for the determination of Cd concentrations in soil samples. They fabricated a transparent multilayer forked-finger Au electrode structure with two sensors and measured the capacitance by the impedance analyzer, obtaining results showing an increase in capacitance with increasing Cd concentration in the soil sample. The platform can be used in conjunction with the low-power system of Vasiljevic et al. [83] for use in the field, which is powered in real-time with a solar charger and integrates the measured data display into a cloud-based system for remote access to the data.

### 3.3. Potentiometric Method

Potentiometry enables the highly selective quantitative determination of heavy metal ions in water, a technique that focuses on measuring the electric potential at zero current. Potentiometry also focuses on the quantitative analysis of ions in solution using selective electrodes [81]. You et al. [84] developed a miniature ion-selective electrode array (μISE) that enables the detection of multiple heavy metal ions on the same chip by integrating four or more ion-selective electrodes. In their study, the micron-scale device array, produced by a microfabrication process, exhibited the advantages of high sensitivity, stability, and short response times. A sodium acetate buffer solution (0.1 M, pH 4.6) with various heavy metal ion concentrations was first introduced at room temperature, and then the potential between the Ag/AgCl reference electrode and the corresponding μISE electrode was measured with a digital multimeter. According to the principle of the Nernst effect, the potential of the reference electrode remains constant, while that of the ion-selective electrode varies with the concentration of the detected ions. The detection limits of Pb^2+^, Cd^2+^, and Hg^2+^ can reach 1, 3, and 1 ppb, respectively, and can be used for testing drinking water quality.

### 3.4. Electrical Conductivity Method

Conductivity testing has become an almost universal detection technique with high sensitivity and selectivity when detecting metals [85]. Among these, capacitively-coupled non-contact conductivity detection (C^4^D) [86,87] has clear advantages, including good robustness, protection against electrode contamination, and a simple manufacturing process.

The CE-C^4^D (capillary electrophoresis–capacitively-coupled non-contact conductivity detection) microchip system provides effective separation and sensitive detection of heavy metals. Liu et al. [88] designed an integrated lock-in amplifier-based detection circuit (Figure 6a) for non-contact conductivity determination of Mn^2+^, Co^2+^, Pb^2+^, Cd^2+^, and Cu^2+^ in heavy metals, which was able to complete the separation and detection within 100 s. The excitation signal from the sinusoidal generator is applied to one electrode, and the cell current it generates is collected at the second electrode. The current is converted to a voltage by an operational amplifier. This signal is then multiplied by a reference signal derived from a phase shifter, which serves the purpose of compensating for the phase shifts in the measured signal. Finally, the output signal is programmed for data acquisition using Labview. In the electrophoresis program, two high-voltage modules are employed to provide the injection and separation voltages. The injection is performed by applying a voltage of 600 V for 10 s along the injection channel, and separation is achieved using a field strength of 300 V/cm. The detection limits for these five heavy metals ranged from 0.7 to 5.4 μM. In the same vein, Petkovic et al. [89] developed an integrated portable device that incorporates a polymer microchip system (Figure 6b), a non-contact conductivity detector, a data acquisition and signal processing system, and a graphical/user interface. The device can detect metal ions in water samples in situ with a detection limit of 5 μM.

### 3.5. Capacitance Method

Capacitive sensing is becoming an increasingly popular technology, as it is simpler to design, more comfortable and practical to operate, less expensive than other methods, and therefore has the potential for mass production [92]. Parallel plate capacitance is one of the simple forms of capacitance where the only factor affecting the capacitance value is the dielectric constant of the object to be measured when the cross-sectional area and spacing between the two plates is set to a constant and is the basis for the application of parallel plate capacitors for detection [93]. Prijo et al. [90] proposed and investigated a portable, low-cost parallel plate capacitance sensor (Figure 6c) for the detection of heavy and alkali metal ion concentrations in solution. The output voltage of the sensor increased linearly for concentration variations from 0 to 10 ppm, with detection limits reaching 0.4 ppm (KCl solution). A thin and long-sided parallel PCB board allows for the measurement of the dielectric response of each target sample using capacitive sensing detection techniques. The system is simple, inexpensive, and portable in design.

With the rapid development of wireless communication technology, wireless sensors based on capacitive induction (LC) resonant circuits are rapidly progressing in the analysis of heavy metal ions in liquids. Liang et al. [91] proposed a wireless microfluidic sensor based on LTCC technology for the real-time detection of metal ions in water. The wireless sensor consists of a planar spiral inductor and a parallel plate capacitor (LC) resonant antenna. The antenna is connected to the vector network analyzer to obtain amplitude S11 parameters, and a laptop records the data of the vector network analyzer in real-time when the tested liquid is evenly pumped into the microchannel of the sensor using a peristaltic pump (Figure 6d). The concentration of metal ions in water can be tested by the S11 and resonant frequency (fr) of the reflection coefficient of the wireless microfluidic sensor. The wireless microfluidic sensor can have detection limits as low as 5 μM for selected metal ion solutions and is expected to be used for fast and convenient detection of heavy metal ion contaminants in industrial wastewater.

## 4. Other Types of Sensors

In addition to optical and electrochemical detection methods, trace heavy metal ion detection can also be achieved based on principles such as mass change as a basis for constructing portable sensing of heavy metal ions. A quartz crystal microbalance (QCM) is a timing device that measures the change in mass per unit area by measuring the change in frequency of a quartz crystal resonator (QCR). QCM sensors operating on the basis of the piezoelectric effect have the advantages of real-time operation, responsiveness, low cost, and application as small portable instruments [94].

Rotake et al. [95] proposed a QCM sensor based on the measurement technique of a portable vector network analyzer (VNA) (Figure 7), which is portable, reproducible, operational with a small sample size and has a simple process, and is suitable for in situ detection. Rapid and selective detection of Hg^2+^ ions was possible using a quartz crystal microbalance (QCM) functionalized with homocysteine (HCys) and cross-linked pyridine dicarboxylic acid (PDCA) with a LOD of 0.1 ppb and a linear detection range of 0.498 to 6.74 mM. Sartore et al. [96] developed a flow-through QCM chemosensor for monitoring trace metal ions in aqueous solutions. The sensor is based on the surface chelation of metal ions on a multifunctional polymer-modified Au electrode using a 9 MHz AT-cut quartz resonator to obtain a surface-modified Au electrode with a strong heavy metal ion complexation capability. The QCM sensor selectively adsorbs trace metal ions such as Cr, Pb, Cu, and Cd by complexing with functional groups present in the polymer from 0.01–1000 ppm in the concentration range of the solution adsorbed. 

In short, the development of various portable sensing technologies has contributed to the success of on-site detection of heavy metal ions. We have listed the comparison of detection performance (Table 1) and the advantages and disadvantages of each method (Table 2) separately in order to provide a reference for routine testing.

## 5. Summary and Outlook

This paper reviewed the optical and electrochemical methods used in recent years for the portable detection of trace heavy metal ions in the field. The sensors constructed by these methods all have the advantages of being small and highly integrated, with extremely high sensitivity and selectivity for heavy metal ions. These portable sensors are simple to operate, do not require cumbersome pre-processing steps, and are cost-effective and versatile. The combination of these sensors with smart devices such as mobile phones, portable photometers, portable Raman meters, and portable vector network analyzers demonstrates good detection capabilities, rapid acquisition of accurate data in real-time, simplified detection procedures, and a high degree of practicality. However, when applied to field testing in different areas (wastewater, seawater, biological samples, etc.), the difference in pH value and water quality may still lead to different results, so there is a lack of reproducibility. In addition, current detection methods are susceptible to other ions in the environment and have poor selectivity. Complex microelectrode manufacturing techniques also limit the development of detection. We can improve in the following areas: (1) Portable devices can be combined with nanomaterials to detect heavy metal ions. Nanomaterials have the characteristics of low detection limit and biocompatibility, and the modified nano-modified materials can be combined with various substrates, have more and larger active centers, improve the reproducibility and sensitivity of detection, and adapt to detection under various complex matrices. (2) The use of biological components (small molecule ligands or biological macromolecules) that can achieve specific recognition is expected to solve the defect of poor selectivity in on-site detection. (3) Research and develop new microelectrode manufacturing technology, reduce the cost of microelectrode preparation, give full play to the advantages of high mass transfer efficiency, fast signal response, and easy integration of microelectrode sensing, and prepare portable and rapid detection microfluidic devices. We believe that with the solution of these problems, breakthroughs will be made in the field of on-site detection of trace heavy metal ions, and portable and reliable ion sensing will be applied to all aspects of the ecological environment.

## Figures and Tables

**Figure 1 sensors-23-04125-f001:**
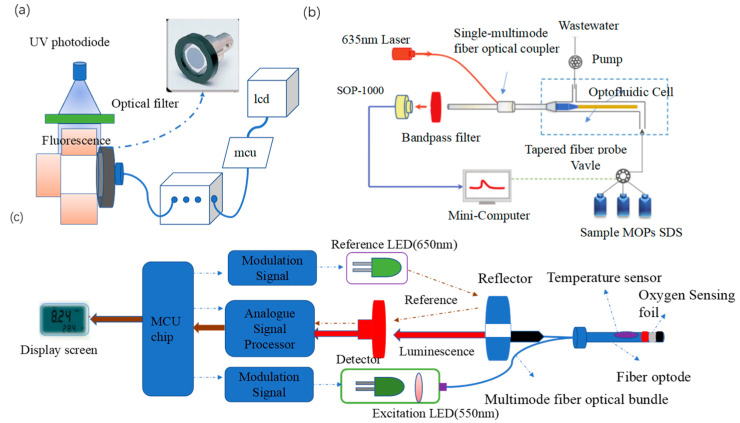
(**a**) Fluorescent quantum dot portable Ag^+^ sensing device [43] (**b**) portable evanescent wave optical fluid Hg^2+^ sensor [44], and (**c**) handheld fiber optic dissolved oxygen sensor [45].

**Figure 2 sensors-23-04125-f002:**
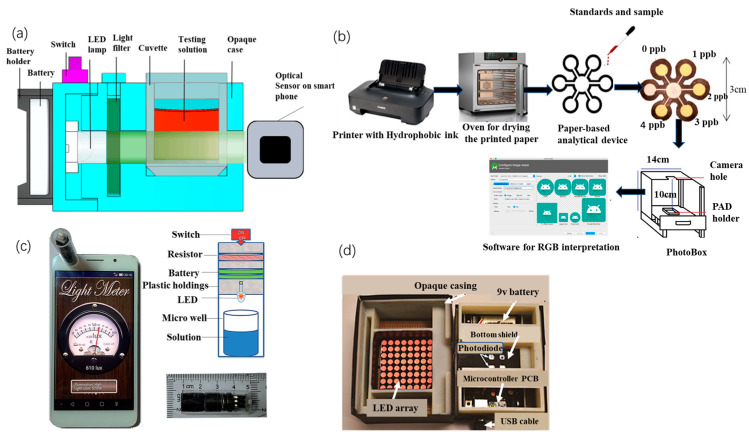
(**a**) Internal structure of the smartphone-based sensing device (SPSD) [51], (**b**) schematic diagram of the preparation and digital image acquisition of a paper-based analysis device (PAD) for the determination of Hg^2+^ [52], (**c**) combined smartphone and ELISA assay [53], and (**d**) inside the colorimetric reader [54].

**Figure 3 sensors-23-04125-f003:**
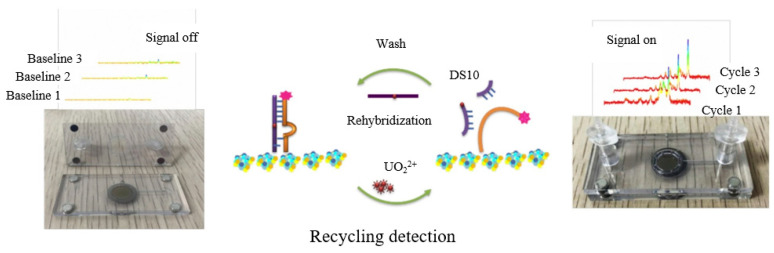
Ultra-sensitive recyclable SERS microfluidic biosensor for the detection of UO_2_^2+^ [62].

**Figure 4 sensors-23-04125-f004:**
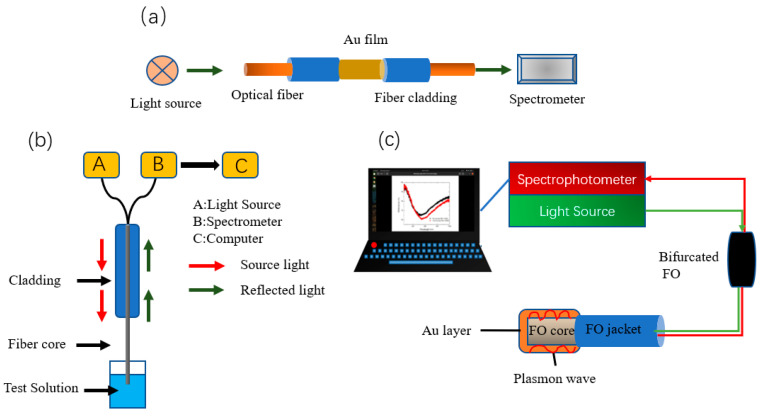
(**a**) Fiber optic SPR sensor detection equipment and process [67], (**b**) Schematic of the experimental setup used for the characterization of the AuNP coated fiber [68], and (**c**) Schematic diagram of the portable FO-SPR sensing system [69].

**Figure 5 sensors-23-04125-f005:**
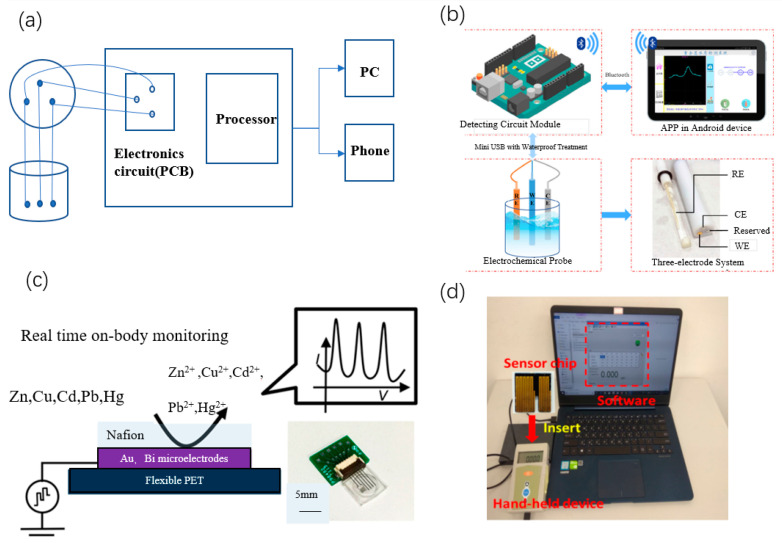
(**a**) Electrochemical sensor for Cd^2+^ and Cu^2+^ detection in soil [74], (**b**) Cu^2+^ portable electrochemical sensing system [75], (**c**) image of microsensor array with a system detection process [76], and (**d**) handheld measurement setup based on impedance changes [77].

**Figure 6 sensors-23-04125-f006:**
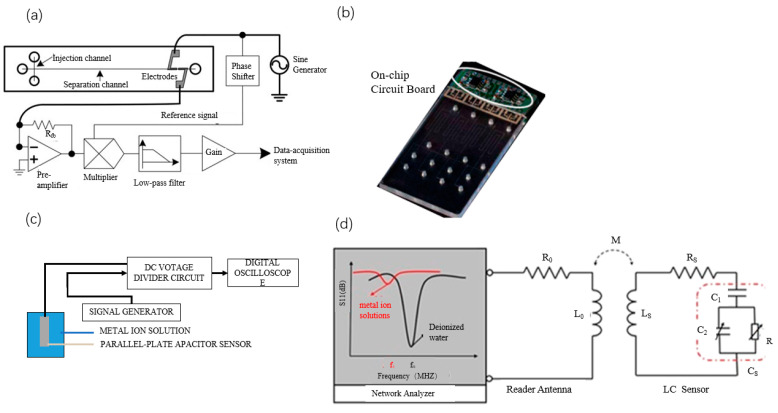
(**a**) CE microchip and detection circuit [88], (**b**) four-channel microchip and non-contact conductivity detector [89], (**c**) experimental setup of a parallel plate capacitive sensor [90], and (**d**) wireless microfluidic sensor based on LTCC technology [91].

**Figure 7 sensors-23-04125-f007:**
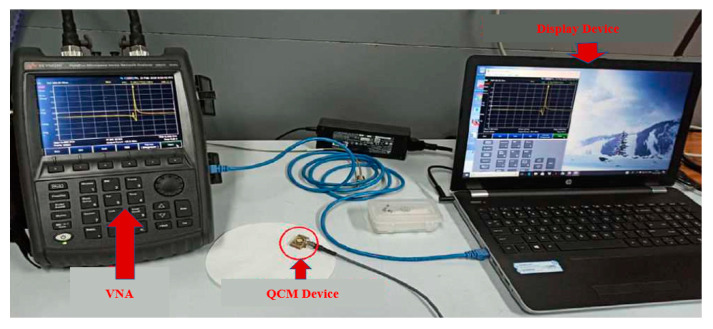
Microfluidic experimental platform for the selective detection of Hg^2+^ ions by a VNA-based QCM sensor [95].

**Table 1 sensors-23-04125-t001:** Detection performance based on different principles.

Principle	Type	Transducer	Detection Range	LOD	Analytes	Ref.
Optical sensors	Fluorescence	TGA/CdTe	5–200 nM	5 nM	Ag^+^	[43]
		CY-A14/BQ-T14	0–400 nM	8.5 nM	Hg^2+^	[44]
		*E. coli*/PtTFPP	0–80 μM	5.62 μM	Hg^2+^	[45]
	Colorimetric	SPSD	0.2–7 μg/mL 5–30 μg/mL 0.1–1 μg/mL 0.05–1 μg/mL	2.18 × 10^−2^ μg/mL 1 μg/mL 0.001 μg/mL 0.02 μg/mL	Pb^2+^ Hg^2+^ Cd^2+^ Zn^2+^	[51]
		AgNPs/Smartphone	5–20 nM	10 nM	Hg^2+^	[52]
		ELISA/Smartphone	0.8–50 ng/mL	0.81 ng/mL	Cr^3+^	[53]
		AgNPs/Chip lab	0–500 ppb 100–400 ppb	30 ppb 89 ppb	Pb^2+^ Al^3+^	[54]
	Raman scattering	Rhodamine B/DNA	1 nM–0.1 μM	7.2 × 10^−7^ μM	UO_2_^2+^	[62]
		I^−^	0.1–10^3^ nM	1 fM	Hg^2+^	[63]
	LSPR/SPR	AuNPs/4-MPY	8–100 nM	8 nM	Hg^2+^	[67]
		AuNPs/MUA	0–100 mM	800 μM	Pb^2+^	[68]
		BSA/Chitosan/PANI	0–100 μM	7.1 nM	Cd^2+^	[69]
Electrochemical sensors	Voltammetry	SWV/ARM	0–400 μg/L	0.0075 μA/μgL^−1^	Cu^2+^	[75]
		SWASV	0–300 μg/L	200 μg/L	Cu^2+^	[76]
	Impedance	ISM-FET	10–100 nM	10^−5^ μM	Pb^2+^	[77]
		α-MnO_2_/GQD	0.001–1 nM	0.81 nM	Pb^2+^	[82]
	Potentiometric	μISE/Ag/AgCl	-	1 ppb 3 ppb 1 ppb	Pb^2+^ Cd^2+^ Hg^2+^	[84]
	Electrical conductivity	CE-C^4^D	0–1 mM	0.7 μM 2.5 μM 5.4 μM 3.5 μM 1.9 μM	Mn^2+^ Cd^2+^ Co^2+^ Cu^2+^ Pb^2+^	[88]
	Capacitance	LTCC	0–5 mM	5 μM	Pb^2+^, Cd^2+^	[91]
Other sensors	QCM	VNA	0.498–6.74 mM	0.1 ppb	Hg^2+^	[95]
		Au electrodes	0.01–10^3^ ppm	-	Pb^2+^	[96]

**Table 2 sensors-23-04125-t002:** Pros and cons of different methods of sensors.

Methods	Pros	Cons
Optical sensors	Fast, high sensitivity, and high resolution	Easily disturbed and complex device composition
Electrochemical sensors	Low cost, simple operation, and high sensitivity	Poor reproducibility and stability
QCM	Repeatable, low sample volume, and real-time monitoring	Low portability

## Data Availability

The data presented in this study are not publicly available at this time but may be obtained upon reasonable request from the authors.

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
