# Peer review of "Advances in Portable Heavy Metal Ion Sensors"

_sensors, 2023, doi:10.3390/s23084125_

Round 1

Reviewer 1 Report

In the present manuscript, authors have reviewed the advances in portable sensors for heavy metal ion detection. The subject is interesting and a comprehensive review of the relevant literature has been covered.   The review paper may be accepted after few minor changes.

 1) Word “progress” should be removed from the key words.

2)  Lines 28-30: sentence “Today, an increasing number ……. through the food chain.” is not clear. It needs to be revised.

3) Remove comma after reference 7 in line 32.

4)  A comparison of detection limits of different methods in tabular form should be given.

Author Response

Thank you for your letter and for the reviewer’s comments concerning our manuscript entitled “Advances in portable heavy metal ion sensors”. The comments are all valuable and very helpful for revising and improving our paper, as well as the important guiding significance to researchers. We have studied comments carefully and have made correction which we hope meet with approval. The main corrections in the paper and the responds to the reviewer’s comments are presented in the list of point-by-point responses. Please see the attachment.

We appreciate for Editor/Reviewer’s warm work earnestly, and hope that the manuscript has been improved satisfactorily and will be accepted for publication.

Once again, thank you very much for your comments and suggestions.

Sincerely yours

Zhengchun Liu, PhD, Prof.

Hunan Key Laboratory for Super Microstructure and Ultrafast Process

School of Physics and Electronic

Central South University

Changsha, 410083, China

MP:015974251534

Email: liuzhengchunseu@126.com

Reviewer 2 Report

This paper details Advances in portable heavy metal ion sensors.

It is an interesting and complete review, many types of sensors for metals being presented in detail.

This paper can be published in Sensors, without other modifications.

Author Response

We appreciate your review and your recognition of my work. We hope to have the opportunity to communicate and learn from each other

Reviewer 3 Report

This review summarize some of the recent advances in the field of portable heavy metal ion sensors. In all, the manuscript is well organized, and the information included is well presented. Certain level of revision is still needed.

1. some sentences are too long and hard to get the point, like lines 43-46, please optimize them. 

2. Lines 54-71, are these information from literature or author? If they came from papers, please insert references. 

3. The units used in the manuscript are not uniform, such as line 0–12 ng-mL-1 in  line 99, ng/mL in lines 177, mmolL-1 in line 223, also please correct the spell in line 324, should be 10-6 M instead of 10 M-6.

4. All the advances in the manuscript are reported individually based on detecting methods. Can they be summarized in a table according to method,  metal ions and detection limits? Also, advantages and disadvantages are welcome to add to compare.

5. In the summary, author mentioned the stability needs to be improved, while in the manuscript, the stability issue are not present in detail. 

6. The section on outlook is usually the most valuable in a high-quality review, but I think it is superficial in this perspective. More detailed suggestions are needed.

Author Response

Thank you for your letter and for the reviewer’s comments concerning our manuscript entitled “Advances in portable heavy metal ion sensors”. The comments are all valuable and very helpful for revising and improving our paper, as well as the important guiding significance to researchers. We have studied comments carefully and have made correction which we hope meet with approval. The main corrections in the paper and the responds to the reviewer’s comments are presented in the list of point-by-point responses.

We appreciate for Editor/Reviewer’s warm work earnestly, and hope that the manuscript has been improved satisfactorily and will be accepted for publication.

Once again, thank you very much for your comments and suggestions.

Sincerely yours

Zhengchun Liu, PhD, Prof.

Hunan Key Laboratory for Super Microstructure and Ultrafast Process

School of Physics and Electronic

Central South University

Changsha, 410083, China

MP:015974251534

Email: liuzhengchunseu@126.com
